# A male-specific *doublesex* isoform reveals an evolutionary pathway of sexual development via distinct alternative splicing mechanisms

Caihong Han[1,3], Qionglin Peng[1,3], Xiangbin Su[1], Limin Xing[1], Xiaoxiao Ji[1] & Yufeng Pan [1,2 ✉]

The *doublesex/mab-3* related transcription factor (*Dmrt*) genes regulate sexual development in metazoans. Studies of the *doublesex* (*dsx*) gene in insects, in particular *Drosophila melanogaster*, reveal that alternative splicing of *dsx* generates sex-specific Dsx isoforms underlying sexual differentiation. Such a splicing-based mechanism underlying sex-specific *Dmrt* function is thought to be evolved from a transcription-based mechanism used in non-insect species, but how such transition occurs during evolution is not known. Here we identified a male-specific *dsx* transcript (*dsx^{M2}*) through intron retention (IR), in addition to previously identified *dsx^M* and *dsx^F* transcripts through alternative polyadenylation (APA) with mutually exclusive exons. We found that Dsx^{M2} had similarly masculinizing function as Dsx^M. We also found that the IR-based mechanism generating sex-specific *dsx* transcripts was conserved from flies to cockroaches. Further analysis of these *dsx* transcripts suggested an evolutionary pathway from sexually monomorphic to sex-specific *dsx* via the sequential use of IR-based and APA-based alternative splicing.

[1] The Key Laboratory of Developmental Genes and Human Disease, School of Life Science and Technology, Southeast University, Nanjing 210096, China. [2] Co-innovation Center of Neuroregeneration, Nantong University, Nantong 226019, China. [3] These authors contributed equally: Caihong Han, Qionglin Peng. ✉email: pany@seu.edu.cn

Sexually dimorphic traits and behaviors are mediated by sex-specific expression of regulatory genes. Substantial studies reveal that the *doublesex/mab-3* related transcription factor (*Dmrt*) genes have been found in all studied animal models and humans for sexual differentiation[1–3]. Dmrt transcription factors share a common DM (Dsx/Mab-3) domain which is a zinc-finger DNA binding motif, but show little sequence conservation in other parts[1]. In humans, *Dmrt1* is linked to disorders of sex development and testicular germ cell tumors[4,5]. *Dmrt1* is preferentially expressed in the male gonad and required postnatally to maintain gonadal sex in mammals and other vertebrates[2,6–8]. In the nematode worm *Caenorhabditis elegans*, *mab-3* is mainly expressed in male tails and necessary for proper morphogenesis and differentiation of copulatory structures[9]. In the fruit fly *Drosophila melanogaster*, sex-specific Dsx isoforms, Dsx^M in males and Dsx^F in females, control sexual differentiation and behaviors in both sexes, and are generated through alternative polyadenylation (APA) with mutually exclusive exons[10–14].

Despite conserved roles of *Dmrt* genes in sexual differentiation in metazoans, regulatory mechanisms underlying their sex-specific function are diversified[1–3,15,16]. The alternative splicing-based regulation of *Dmrt* genes has only been found in insects and often mediated by the female-specific expression of *transformer* (*tra*) with few exceptions[10,16–19]. In vertebrates and nematodes, *Dmrt* genes are not sex-specifically spliced, but transcribed in a predominantly male-specific manner for the development of male-specific traits[2]. It has been proposed that the insect-specific mechanism based on alternative splicing of *dsx* evolved from a more ancient mechanism based on male-specific transcription[16,17], but how such transition occurs during evolution is rarely known.

The canonical insect *tra-dsx* pathway generates sex-specific Dsx isoforms by the selective use of distinct 3′ exons in the two sexes[16,17]. In *D. melanogaster* females, Tra and Tra-2 bind to the female-specific exon of *dsx* and direct female-specific splicing of *dsx^F*, while in males the absence of Tra function permits the default splicing of *dsx^M* including both common and male-specific exons[10,20–22]. Dsx^M and Dsx^F have common DM domain and two oligomerization domains, thus may bind to the same sets of target genes; however, they have been found to oppositely regulate target gene expression to establish male- or female-specific differentiation, possibly through their sex-specific C-terminus[23–26].

In this study, we report that intron retention (IR) generates a male-specific *dsx* transcript and propose an evolutionary pathway from sexually monomorphic to sex-specific splicing of *dsx*. We identify another male-specific *dsx* transcript (*dsx^M2*), in which the intron linking the last common exon and the female-specific exon is retained. The IR-generated Dsx^M2 has a masculinizing role like Dsx^M and is crucial for male courtship robustness in *D. melanogaster*. We further show that such mechanism generating sex-specific *dsx* transcripts is deeply conserved in insects from fruit flies to cockroaches. The male-specific intron retention depends on the presence of a weak splicing acceptor sequence inside the intron, as well as the regulation of Tra. Comparison of the structures of *dsx* transcripts, in light of their difference of conservation, suggests an evolutionary pathway from sexually monomorphic to sex-specific *dsx* via the sequential use of IR-based and APA-based alternative splicing.

## Results

### Identification of a novel *dsx* transcript *dsx^M2* through intron retention.

It has been determined that the *dsx* pre-mRNA is sex-specifically spliced to yield the male-specific *dsx^M* and female-specific *dsx^F* mRNAs. In an attempt to determine relative expression of *dsx* transcripts in the two sexes in wild-type Canton-S (*wtcs*) and *dsx* mutant (*dsx^683-7058/dsx^1649-9625*) flies, we performed quantitative PCR (qPCR) experiments using two pairs of primers targeting the male-specific or female-specific locus in *dsx*, respectively (Fig. 1a). We found that *dsx^M* was indeed expressed exclusively in *wtcs* males, but not in *wtcs* females or *dsx* mutant flies; however, comparable levels of *dsx* expression were detected in both *wtcs* males and females, but not in *dsx* mutant flies, by using the primer pair targeting *dsx^F* (Fig. 1b, c). These results suggest the presence of *dsx^F* or a novel *dsx* transcript, in addition to *dsx^M*, in male flies.

To identify the *dsx* transcript detected in male flies, we next performed reverse transcription PCR (RT-PCR) experiments using three different primer pairs targeting different sites of the *dsx^F* transcript (Fig. 1d), and found that all PCR products from male flies were larger than those from females (Fig. 1e). The same results were also obtained in another broadly used wild-type *w^1118* flies (Fig. 1e). By sequencing these PCR products and sequence alignment, we found that the novel *dsx* transcript in males is quite similar to *dsx^F*, with only 114 bp intron not being spliced out and retained between the last common exon and the female-specific exon (Fig. 1f, g). Thus, we identified a novel male-specific *dsx* transcript, hereafter referred to as *dsx^M2*, in addition to the previously identified male-specific *dsx^M* and female-specific *dsx^F* transcripts (Supplementary Fig. 1a). Indeed, previous RNA-seq results already suggested the possibility of the 114 bp intron retention in males but not females[27,28] (Supplementary Fig. 1b and Supplementary Table 1). The predicted Dsx^M2 protein contains the common N-terminus of Dsx proteins and a specific C-terminus with only six amino acids resulting from the early stop codon inside the retained intron (Supplementary Fig. 2).

### *dsx^M2* functions like *dsx^M* but has limited roles.

The above results identified a male-specific *dsx^M2* transcript, but how it is generated from intron retention and whether it plays any role in sexual development or behavior is not known. To investigate whether and where *dsx^M2* expresses, we tried to generate a polyclonal antibody against the eight amino acids in the C-terminus of the predicted Dsx^M2 protein but failed to detect any signal through immunostaining and western blot experiments. We next performed qPCR experiments to quantify the relative expression levels of *dsx^M* and *dsx^M2* in various tissues. We found that *dsx^M2* was expressed in the fly head, thorax, and forelegs, but the level of *dsx^M2* mRNA was generally lower than the level of *dsx^M* mRNA (Supplementary Fig. 3), which is also consistent with the result using the whole fly body (Fig. 1b). To investigate whether *dsx^M2* plays a role in sexual development and/or behavior, we generated a *UAS-dsx^M2RNAi* construct targeting the retained intron (Fig. 2a), combined with previously used *UAS-dsx^MRNAi* construct targeting the male-specific exon[14], and validated their efficiency with three different *GAL4* drivers, the *actin-GAL4*, the pan-neuronal *R57C10-GAL4*[29], and *dsx^GAL4*, using qPCR (Fig. 2b, c). Both *dsx^M* and *dsx^M2* RNAi lines knocked down the corresponding *dsx* mRNA efficiently though not in the same level (Fig. 2b, c), and did not reduce the level of the other transcript (Supplementary Fig. 4). We next knocked down *dsx^M* or *dsx^M2* in all *dsx*-expressing cells [*dsx^GAL4(Δ2)*] and found that males with *dsx^M* knocked down were intersexual and displayed little courtship, while males with *dsx^M2* knocked down had regular male appearance but reduced courtship and mating success with virgin female targets (Fig. 2d). These results suggest that *dsx^M2* is involved in regulating male courtship but not, if any, development of sexually dimorphic traits. To further confirm whether *dsx^M2* functions in the nervous system to mediate male courtship, we knocked down *dsx^M* or *dsx^M2* pan-neuronally using

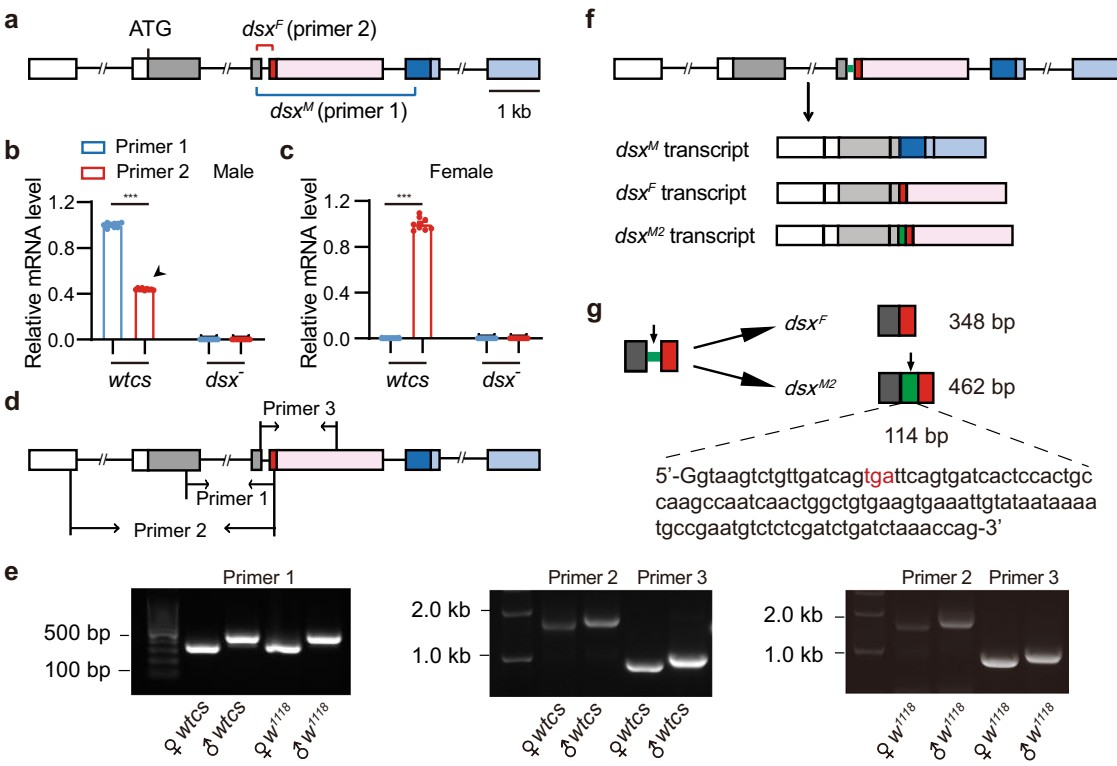

**Fig. 1 Identification of a male-specific *dsx^M2* transcript through intron retention. a** Schematic structure of the *dsx* gene. Boxes: exons; white and gray boxes: common 5'-UTR and coding sequences respectively in both sexes; red and pink: coding and 3'-UTR sequences of the female-specific exon; dark and light blue: coding and 3'-UTR sequences of male-specific exons. Primers against sex-specific sequences are indicated. **b**, **c** Relative mRNA expression levels of presumed *dsx^M* cycle at 60% humiditytranscript (**b**, primer 1) and *dsx^F* transcript (**c**, primer 2) in *wtcs* and *dsx* mutant (*dsx^683-7058*/*dsx^1649-9625*) males and females. *n* = 9 based on three replicates for each. \*\*\**p* < 0.001, Mann–Whitney U test. Note that a *dsx* transcript (arrowhead) is detected in males using the primer against the female-specific transcript (**b**). **d**, **e** Reverse transcription PCR (RT-PCR), using three pairs of primers as indicated (**d**), and sequencing identify the male-specific *dsx^M2* transcript, in addition to the female-specific *dsx^F* transcript, in both *wtcs* and *w^1118* flies (**e**). For primer 1: 462bp and 348bp for male and female products, respectively; for primer 2: 1833bp and 1719bp for male and female products, respectively; for primer 3: 1069bp and 955bp for male and female products, respectively. **f** Illustration of the male-specific *dsx^M2* transcript, in addition to previously identified *dsx^M* and *dsx^F* transcripts. **g** The *dsx^M2* transcript differs from the *dsx^F* transcript with the 114 bp intron (in green) retained. Red letters (tga) indicate the stop codon inside the retained intron.

the *R57C10-GAL4* driver. Males with *dsx^M* or *dsx^M2* knocked down pan-neuronally had regular male appearance but much reduced courtship levels and mating success (Fig. 2e). These results indicate a crucial role of *dsx^M2*, like *dsx^M*, in regulating male courtship intensity.

To further investigate *dsx^M2* function, we generated constructs overexpressing each *dsx* isoform (*UAS-flag-dsx^F*, *UAS-myc-dsx^M* and *UAS-flag-dsx^M2*) and validated their efficiency using qPCR experiments and immunostaining with anti-Flag and anti-Myc antibodies (Supplementary Fig. 5). We then overexpressed these *dsx* isoforms in all *dsx*-expressing cells and found that over-expressing Dsx^F feminized male development (genitals and sex combs), while overexpressing Dsx^M or Dsx^M2 masculinized female development (genitals) (Fig. 2f). To avoid potential interference of overexpressed and indigenous Dsx isoforms (*e.g.*, Dsx^M2 and Dsx^F), we next overexpressed these *dsx* isoforms in a *dsx* mutant background [*dsx^GAL4(Δ2)*/*dsx^1649-9625*] in which both sexes were intersexual, and found that expressing Dsx^F induced female differentiation in both sexes, while expressing Dsx^M or Dsx^M2 strongly masculinized genital and sex comb development in both sexes (Fig. 2g). These results indicate that Dsx^M2 has a potential masculinizing role like Dsx^M.

As transcription factors, Dsx proteins regulate sexual differ-entiation through their target genes, of which three genes encoding the female-specific Yolk Proteins (YPs) have been

intensively studied under control of Dsx[10,24,30]. To compare the transcriptional regulation of different Dsx isoforms on target genes, we overexpressed the three Dsx isoforms driven by the *dsx^GAL4(Δ2)* and tested relative expression changes of *yp2* and *yp3*. Compared to control females, overexpressing Dsx^F significantly increased *yp2* and *yp3* expression, while overexpressing Dsx^M or Dsx^M2 significantly reduced their expression in females (Fig. 2h). Note that *yp2* and *yp3* expression were more severely reduced in females expressing Dsx^M than those expressing Dsx^M2, suggesting that Dsx^M2 had a weaker role of transcriptional inhibition compared to Dsx^M (Fig. 2h). The expression of *yp2* or *yp3* was undetectable in control males, but significantly increased in males overexpressing Dsx^F, which further confirmed the role of Dsx^F in promoting *yp2* and *yp3* transcription (Fig. 2i). Taken together, the knockdown experiments indicate a crucial role of *dsx^M2* in the nervous system for male courtship robustness but not sexual development, while the overexpression experiments indicate that Dsx^M2 has a potentially masculinizing role like Dsx^M.

**Intron retention is a common mechanism to generate sex-specific *dsx* isoforms.** As we identified the male-specific *dsx^M2* transcript through intron retention, which is a potent mechanism underlying sexual differentiation, we asked if such an alternative splicing form also existed in other animal species. We tested three

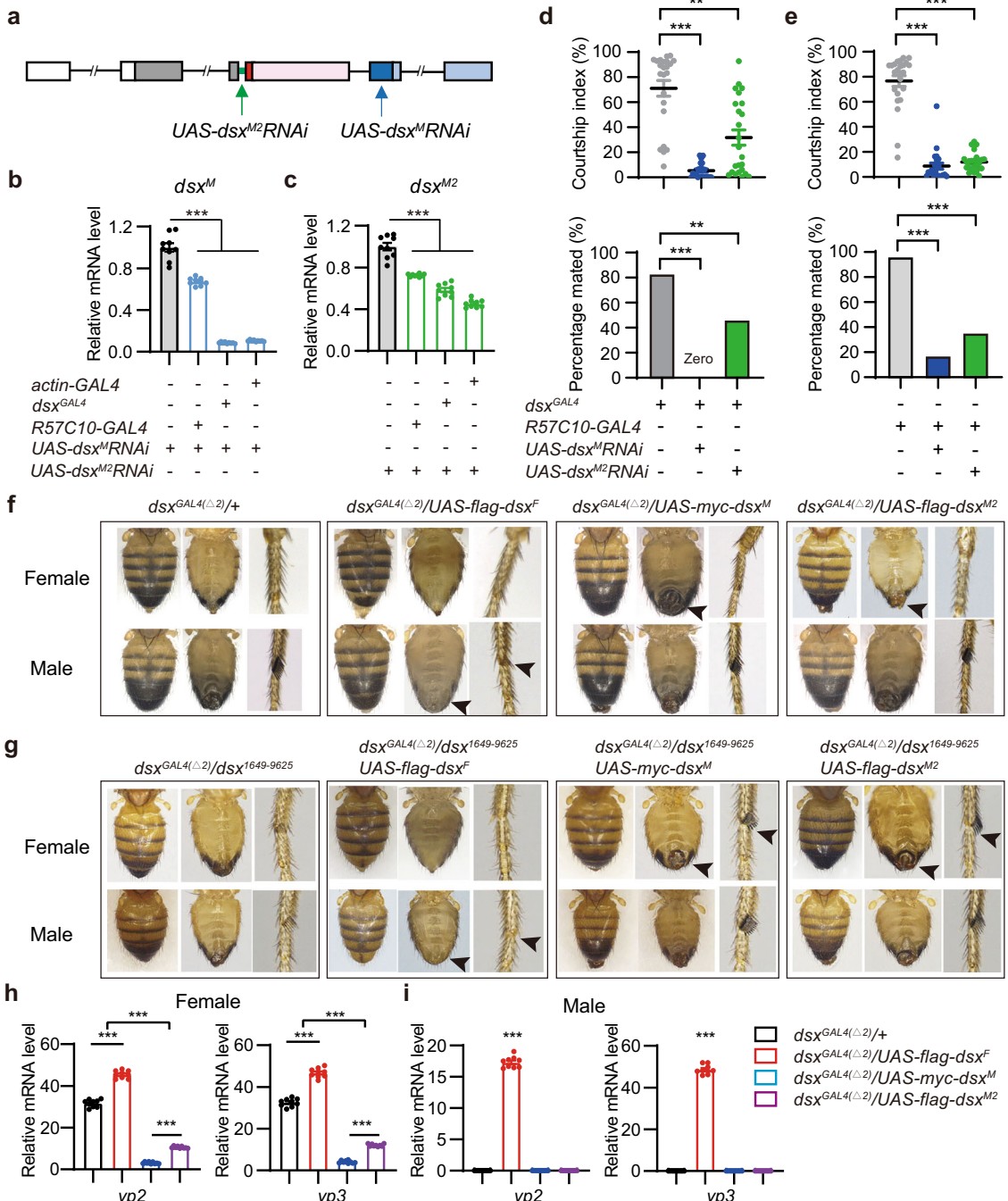

**Fig. 2 dsx^M2 has a potentially masculinizing function like dsx^M. a** RNAi targeting *dsx^M* and *dsx^M2* as indicated by blue and green arrows, respectively.
**b**, **c** Relative mRNA expression levels of *dsx^M* (**b**) and *dsx^M2* (**c**) in control and RNAi-mediated males. *n* = 9 based on three replicates for each. ***p* < 0.001,
Mann-Whitney U test. **d**, **e** Knocking down *dsx^M* or *dsx^M2* in *dsx*-expressing cells (**d**) or pan-neuronally (**e**) impairs male courtship and mating success with
females. *n* = 23, 24, 24, 24, 24, 24 for *dsx^GAL4/+*, *dsx^GAL4/UAS-dsx^MRNAi*, *dsx^GAL4/UAS-dsx^M2RNAi*, *R57C10-GAL4/+*, *R57C10-GAL4/UAS-dsx^MRNAi* and
*R57C10-GAL4/UAS-dsx^M2RNAi* respectively. For courtship index, ***p* = 0.0012 and ****p* < 0.001, Kruskal-Wallis test with Dunn's multiple comparisons test;
for percentage mated, ***p* = 0.0087 and ****p* < 0.001, Chi-square test. **f**, **g** Overexpression of Dsx^F feminized, while overexpression of Dsx^M or Dsx^M2
masculinized *dsx*-expressing cells, including external genitalia and sex comb (arrowhead), under wild-type background [**f**, *dsx^GAL4(Δ2)/+*] or mutant
background [**g**, *dsx^GAL4(Δ2)/dsx^1649-9625*] for *dsx*. **h**, **i** Relative mRNA expression levels of *dsx* target genes, *yp2* and *yp3*, in females (**h**) and males (**i**).
Expression levels of *yp2* and *yp3* were increased with Dsx^F overexpression and decreased with Dsx^M or Dsx^M2 overexpression. *n* = 9 based on three
replicates for each. ****p* < 0.001. Mann-Whitney U test. Error bars indicate SEM.

other *Drosophila* species including the closely related *D. simulans* and two other distant species *D. mojavensis* and *D. virilis*, by performing RT-PCR experiments using multiple primer pairs targeting sequences including the potentially retained intron and

a portion of the female-specific exon, followed by sequencing. We found retention of the 118 bp intron in *D. simulans* males (Fig. 3a–c), 131 bp intron in *D. mojavensis* males (Fig. 3d–f) and 109 bp intron in *D. virilis* males (Fig. 3g–i), which generated

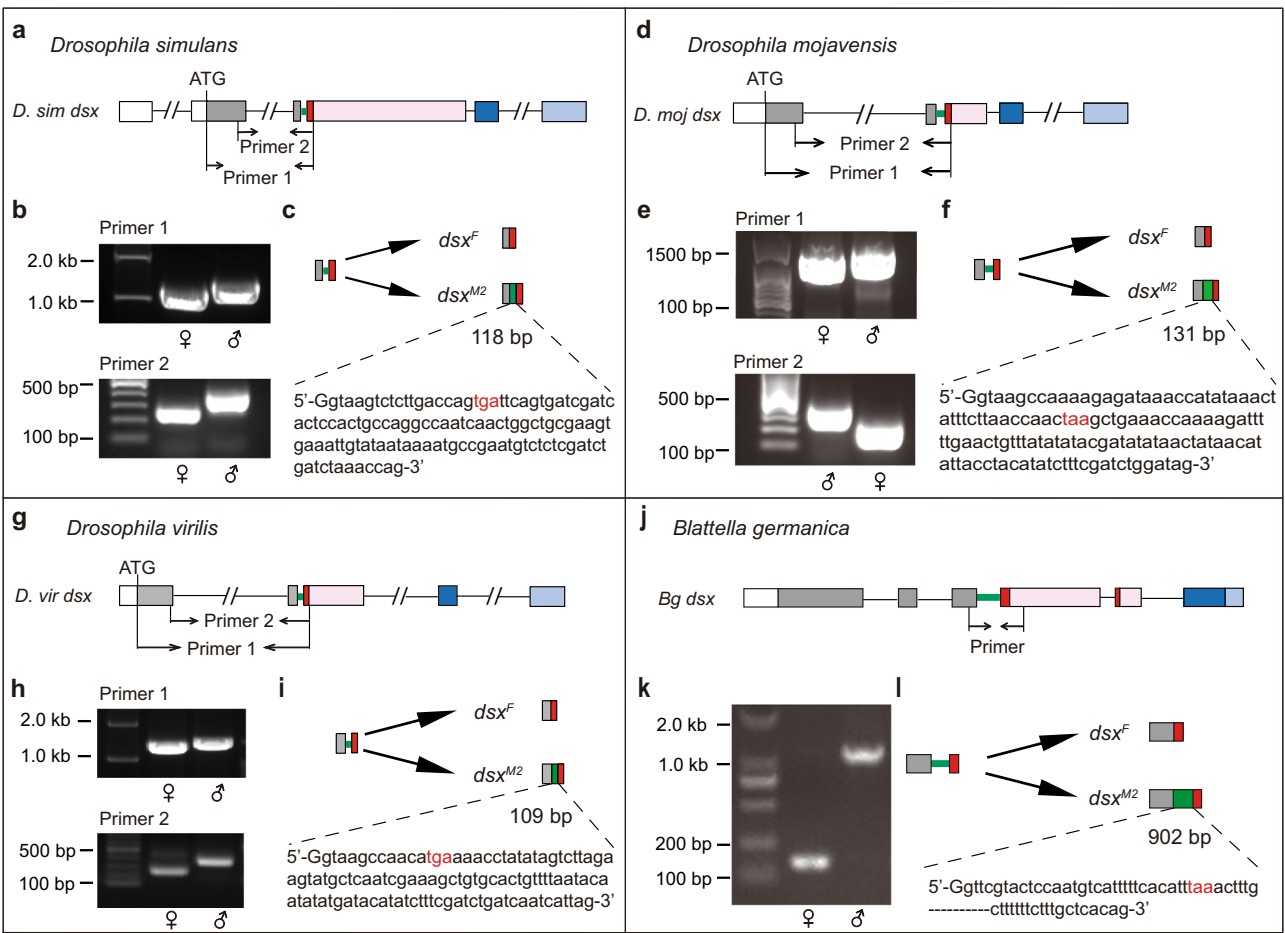

**Fig. 3 Intron retention-induced sex-specific splicing is widely conserved.** Retention of the intron that separates the last common exon and the female-specific exon generates male-specific *dsx*^M2^ transcripts in *D. simulans* (**a–c**), *D. mojavensis* (**d–f**), *D. virilis* (**g–i**) and *B. germanica* (**j–l**). The *dsx* gene structures are illustrated in **a**, **d**, **g** and **j**. White and gray boxes indicate common 5′-UTR and coding sequences respectively in both sexes. Red and pink boxes indicate coding and 3′-UTR sequences of the female-specific exon. Dark and light blue boxes indicate coding and 3′-UTR sequences of male-specific exons. The thick green line indicates the intron spliced out in the *dsx*^F^ transcript and retained in the *dsx*^M2^ transcript. Primers for RT-PCR experiments (**b**, **e**, **h** and **k**) are indicated respectively (**a**, **d**, **g** and **j**). Full sequences of the retained intron of *dsx*^M2^ are indicated (**c**, **f** and **i**) except for the 902bp one in *B. germanica dsx*^M2^ (**l**). The capital letter "G" from the upstream exon is added in front of the intron sequence to ensure that the stop codon highlighted in red is aligned within the open reading frame.

*dsx*^M2^ transcripts in the same way as in *D. melanogaster* males. All predicted Dsx^M2^ proteins have conserved DM domain including a zinc-finger DNA binding domain and oligomerization domains, and few amino acids in the C-terminus due to stop codons in the retained intron (Supplementary Fig. 6). Protein sequence alignment of Dsx isoforms in these *Drosophila* species revealed that sex-specific C-terminus of Dsx^F^ proteins were almost identical, while those of Dsx^M^ and Dsx^M2^ proteins were less conserved (Supplementary Fig. 7), which is consistent with previous findings on the different levels of conservation of Dsx^M^ and Dsx^F^ isoforms across insect species[31].

To further test whether intron retention could be a conserved way to generate alternative splicing of *dsx*, we performed RT-PCR experiments targeting the *dsx* gene in a hemimetabolous insect, the German cockroach *Blattella germanica*, as it is not only a far distant insect species apart from *D. melanogaster* (>300 million years), but also with the *dsx* gene structure being previously studied[17,32]. We designed a pair of primer targeting the last common exon and the first female-specific exon respectively (Fig. 3j) and obtained sex-specific fragments by PCR (Fig. 3k). Through sequencing, we revealed the existence of the male-specific *dsx*^M2^ transcript that retains the 902 bp intron linking the last common exon and the first female-specific exon (Fig. 3l), in addition to previously identified *dsx* transcripts[17]. Taken together, these results indicate that intron retention is a common mechanism to generate sex-specific *dsx* isoforms in distant insect species.

**dsx intron retention is jointly regulated by a weak splice acceptor and transformer.** To further investigate factors affecting intron retention and generating sex-specific *dsx* isoforms, we compared sequences of all introns of *dsx* in the four *Drosophila* species. We did not compare the acceptor sequences in *Blattella germanica*, as we have insufficient knowledge about the consensus acceptor sequence in this species. We found that all introns of *dsx* in *D. melanogaster* and *D. simulans* start with the donor sequence GTAAGT and end with acceptor sequences (T/C)ₙNCAG (N indicates A, C, G or T) (Fig. 4a, b). Previous studies already suggest such consensus acceptor sequence[10], where the number of pyrimidines (T/C) upstream of the last four nucleotides (NCAG) is at least 9. Indeed, the common (intron 1 and 2) and male-specific (intron 3m and 4m) acceptors have 9–11 pyrimidines upstream of NCAG. As for the retained intron (spliced out in female, 3f), the acceptor has only 6 pyrimidines upstream of NCAG (Fig. 4a, b). We next compared donor and acceptor

| Structure of the *dsx* gene | Donor sites | Acceptor sites | | |
|---|---|---|---|---|
| **a** *D. melanogaster* | 1 GTAAGT------ | ------ AACTTTCTCTTT (10) | T | CAG |
| | 2 GTAAGT------ | ------ TCTCCGTTTATT (10) | C | CAG |
| | 3f GTAAGT------ | ------ ATCTGATCTAAA (6) | C | CAG |
| | 3m | ------ TTCTGTTAATCC (9) | C | CAG |
| | 4m GTAAGT------ | ------ CTTTGTTCTTCC (11) | A | CAG |
| **b** *D. simulans* | 1 GTAAGT------ | ------ AACTTTTTCTTT (10) | T | CAG |
| | 2 GTAAGT------ | ------ TCTCCGTTTATT (10) | T | CAG |
| | 3f GTAAGT------ | ------ ATCTGATCTAAA (6) | C | CAG |
| | 3m | ------ TTCTGTTAATCC (9) | C | CAG |
| | 4m GTAAGT------ | ------ TTTTGTTCGTCC (10) | A | CAG |
| **c** *D. mojavensis* | 1 GTAAGT------ | ------ TTTGTTTTATTT (10) | G | CAG |
| | 2f GTAAGC------ | ------ CTTTCGATCTGG (8) | A | TAG |
| | 2m | ------ TTGTTCGTCTCC (10) | A | CAG |
| | 3m GTAAGT------ | ------ CTCTTCCACACT (10) | T | TAG |
| **d** *D. virilis* | 1 GTAAGT------ | ------ GTTTCTCCCTTT (11) | A | CAG |
| | 2f GTAAGC------ | ------ TCTGATCAATCA (7) | T | TAG |
| | 2m | ------ TCTATTCGTTTC (10) | A | CAG |
| | 3m GTAAGT------ | ------ TTCCACTCGCGT (9) | T | TAG |
| Consensus | GTAAGY------ | ------ YYYYYYYYYYYY (12) | N | YAG |

**e**

```
D. mel   gtaagtctgttgatcagtgattcagtgatc------actccactgccaagccaatcaactggctgtgaagtgaaattgta    74
D. sim   gtaagtctcttgaccagtgattcagtgatcgatcactccactgccaggccaatcaactggctgcgaagtgaaattgta    78
D. moj   gtaagccaaaagagataaaccatataaactatttcttaac-----------caactaagtgaaaccaaaagatttttga-   68
D. vir   gtaagccaacatgaaaacctatatagtcttagaagtatgct----------caatcgacaagctgtgcactgtttactgt-   70
         ***** *                          ************  *  **   ***

D. mel   -------------------------taataaaatgccga----atgtctctcgatctgatctaaaccag   114
D. sim   -------------------------taataaaatgccga----atgtctctcgatctgatctaaaccag   118
D. moj   ttatatatacgatatataactataacatattacctacatatctttcgatctg---------gatag   131
D. vir   -------------------------taatacaatatatgatacatatctttcgatctgatcaatcattag   109
         ***********  **  ** *** ************  *  *  **
```

**Fig. 4 Retained introns share conserved sequence properties. a–d** Comparison of donor and acceptor sequences of spliced and retained introns. The structure of the *dsx* gene including introns (black lines), the retained intron for *dsx*$^{M2}$ (thick green line), and exons (boxes), as well as splicing donor and acceptor sequences for each intron in *D. melanogaster* (**a**), *D. simulans* (**b**), *D. mojavensis* (**c**) and *D. virilis* (**d**). Introns were labeled with numbers followed by "f" (female-specific splicing) or "m" (male-specific splicing). respectively. As for the consensus sequences, Y indicates pyrimidine, and N indicates either A, G, T or C. The number of pyrimidines in the 12 upstream nucleotides of the acceptor sequence NYAG is indicated in the brackets for each intron. **e** Comparison of retained intron sequences of *dsx*$^{M2}$ in four *Drosophila* species. Underlined letters indicate potential Tra binding sites. Black asterisks indicate perfect matches of amino acids among four species, and gray asterisks indicate three matches out of four species.

sequences of introns of the *dsx* gene in *D. mojavensis* and *D. virilis*. We found similar results as in *D. melanogaster* and *D. simulans*: the common and male-specific acceptors have 9–11 pyrimidines, while the female-specific acceptors have 7 or 8 pyrimidines upstream of NCAG (Fig. 4c, d). These results indicate that the presence of a weak splice acceptor (6–8 pyrimidines) may lead to intron retention of the *dsx* gene in males of four *Drosophila* species.

We next compared full sequences of retained introns in the four *Drosophila* species. Surprisingly, these intron sequences, despite many differences, share intensive identity in two regions: one in the 3′ end of the intron overlapping the region of the weak acceptor site that may contribute to intron retention as above mentioned, and the other in the middle of these introns containing the core sequence (CAATCAAC) of the *tra* binding sequence, TC(T/A)(T/A)CAATCAACA, that occurs six times in the female-specific exon (Fig. 4e). The conservation of the above core sequence, which is a potential *tra* binding site within the retained introns, suggests its potential role in *tra*-mediated alternative splicing. Indeed, we observed intron retention in females with *tra* knocked down in all cells (*actin-GAL4/+; UAS-traRNAi/+*) (Supplementary Fig. 8). Together these results suggest that the weak splice acceptor and *tra* regulation jointly determine the splicing modes between *dsx*$^{M2}$ (intron retained) and *dsx*$^{F}$ (intron spliced out).

**A proposed evolutionary pathway of *dsx* alternative splicing.** The identification of the *dsx*$^{M2}$ transcript promoted us to analyze how the two forms of alternative splicing (IR and APA) were evolved to generate the modern day *dsx* transcripts (Fig. 5a). Substantial studies already elucidated an evolutionary pathway from sexually monomorphic *dsx* to sex-specific *dsx* transcripts upon the evolution of *tra-dsx* regulation in females[16,17]. A simple assumption is that sex-specific *dsx* transcripts come from a sexually monomorphic transcript *dsx*$^{M2}$, or *dsx*$^{M}$, or both. Due to the nature of close relationship between *dsx*$^{M2}$ and *dsx*$^{F}$ (whether an intron with a weak splicing acceptor is spliced out or not), and the more conservation of the female-specific exon than the male-specific exons in insect species[31,33], we propose that *dsx*$^{M2}$ could serve as an ancient monomorphic transcript in both sexes, followed by two sequential events evolving alternative splicing: the IR-based mechanism generating *dsx*$^{M2}$ in males and *dsx*$^{F}$ in females with the evolution of Tra regulation in females, and later the APA-based mechanism generating the rapid-changing *dsx*$^{M}$ with the evolution of a stronger splicing acceptor to capture male-specific exons (Fig. 5b).

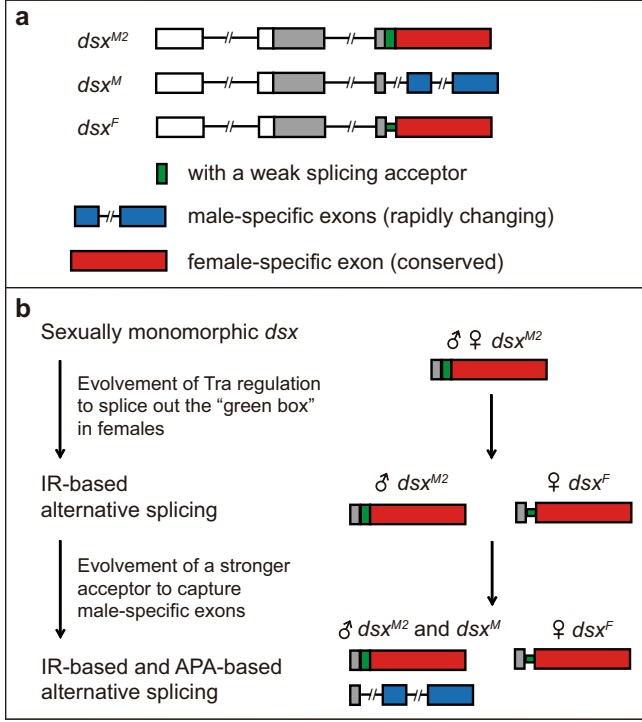

**Fig. 5 An evolutionary pathway of *dsx* alternative splicing underlying sexual development. a** Structures of the modern day *dsx* transcripts, *dsx^M2^*, *dsx^M^*, and *dsx^F^* in *D. melanogaster*. **b** A proposed evolutionary pathway from sexually monomorphic *dsx^M2^* to sex-specific *dsx* transcripts sequentially via the IR-based and APA-based mechanisms.

## Discussion

Previous studies have shown that alternative splicing of *dsx* generates sex-specific isoforms (Dsx^M^ and Dsx^F^) underlying sexual differentiation in *D. melanogaster* and many other insects[1,2,10,16]. Such splicing mechanism involves the selective use of sex-specific 3′ exons, which is under regulation of Tra and Tra-2[10,20–22]. Our results identified the male-specific Dsx^M2^ isoform through intron retention, which is a common mechanism generating sex-specific Dsx isoforms in insects across 300 million years.

Recent studies on the alternative splicing of *dsx* using a variety of insect species revealed that the *tra*-mediated female-specific splicing of *dsx* could be evolved from sexually monomorphic isoforms with male-biased expression[16,17]. Indeed, vertebrates, nematodes and crustaceans use male-biased transcription of *Dmrt* genes to direct male-specific function[1,2,34]. In this regard, the evolution of *dsx^F^* may come from *dsx^M^* and/or *dsx^M2^* under Tra regulation, possibly through the functional gain of Tra binding sites in the female-specific exon of *dsx*. Indeed, knocking down Tra function in females induced intron retention-based splicing of *dsx^M2^* instead of *dsx^F^*. The coding sequence of the female-specific exon, which is also included as 3′-UTR of *dsx^M2^*, is more conserved than the male-specific exon of *dsx^M^* that rapidly changes not only across insect orders but also across closely related genera within each insect order[31,33], excluding the possibility of *dsx^M^* as the origin of sexually monomorphic *dsx* transcript. The identification of the intron retention-based *dsx^M2^* transcript fits well into a position as the ancestral isoform of *dsx* expressed in both sexes before the evolution of *dsx^M^* and *dsx^F^*. The conserved presence of a weak splicing acceptor upstream of the female-specific exon provides the selection basis for later recruitment of Tra regulation to generate the female-specific *dsx^F^* from *dsx^M2^* in females, as well as the evolution of a stronger

splicing acceptor to capture male-specific exons and generate *dsx^M^* in males. Such an evolutionary pathway of generating sex-specific Dsx isoforms could fill the gap between the splicing-based mechanism in insects and transcription-based mechanism in vertebrates and nematodes.

The Dsx^M2^ isoform contains all common Dsx amino acids and six specific C-terminal residues, thus could act like other Dmrt transcription factors. Our knockdown experiments reveal that Dsx^M2^ still functions in the nervous system to promote courtship robustness in *D. melanogaster*, but not in the development of male-specific traits such as genitals and sex combs. In addition, the overexpression experiments indicate that Dsx^M2^ has a potential masculinizing role just like Dsx^M^. However, a direct comparison of Dsx^M^ and Dsx^M2^ function could not be faithfully achieved as knockdown and overexpression levels of *dsx^M^* and *dsx^M2^* may be different. Another caveat of these results is the lack of direct evidence of Dsx^M2^ expression in males. It is also possible that Dsx^M2^ only serves as an ancient Dsx isoform whose function has largely been replaced during evolution (*e.g.*, by Dsx^M^), and now has limited expression and function. Future studies could generate better reagents to testify Dsx^M2^ expression in a temporal and spatial manner in *D. melanogaster* and other insect species to better understand its function as a possible origin of sex-specific Dsx isoforms underlying sexual development and sexual dimorphism.

## Methods

**Fly stocks**. Flies were raised at 22 °C or 25 °C at 12 hr light/12 hr dark cycle at 60% humidity. *Canton-S* (*wtcs*) and *w^1118^* were used as wild-type strains. *dsx* mutant lines used in Fig. 1b and c include *dsx^683-7058^* and *dsx^1649-9625^*, which were used as previously[13]. *Drosophila mojavensis*, *Drosophila simulans* and *Drosophila virilis*[13], *R57C10-GAL4* (attP2, BDSC_39171), *actin-GAL4* (BDSC_25374), *UAS-dsx^M^RNAi* (attP2)[14], *UAS-flag-dsx^F^* (attP40) and *UAS-myc-dsx^M^* (attP40)[35], *dsx^GAL412^* and *dsx^GAL412(Δ2)^*[36] were used as described previously. *UAS-dsx^M2^RNAi* (attP2) and *UAS-flag-dsx^M2^* (attP40) were generated in this study and described below in details. Detailed information about fly stocks and other materials used in this study is listed as Table 1.

**_dsx_ sequences and multiple sequence alignment**. We downloaded *dsx* gene sequence of *D. melanogaster* from FlyBase (http://flybase.org/). We used NCBI (https://www.ncbi.nlm.nih.gov/) to find the location and gene sequences of *dsx* in following species: *D. simulans* (Gene ID: 6727147), *D. mojavensis* (Gene ID: 6574377), *D. virilis* (Gene ID: 6633147) and *B. germanica*[17,32]. Gene sequence or amino acid sequences comparisons were performed using Clustal Omega and LaserGene software.

We also used public RNA-seq data[27] (GSM694258 and GSM694259 for *D. melanogaster* females, GSM694260 and GSM694261 for *D. melanogaster* males, https://www.ncbi.nlm.nih.gov/geo/query/acc.cgi?acc=GSE28078) to analyze splicing events of *dsx* (Supplementary Fig. 1b) using the Integrative Genomics Viewer (IGV)[37].

**Generation of _UAS-dsx^M2^RNAi_ line**. To generate the *UAS-dsx^M2^RNAi* construct, the *UAS-dsx^F^RNAi* plasmid (based on the pVALIUM20)[14] was digested to be linearized with *NheI* and *EcoRI*. We selected a 21 nucleotides sequence (tggctgtgaagtgaaattgta) targeting the 114 bp intron, which was synthesized and generated into hairpin by annealing. The resulting DNA fragment and linearized vector were ligated. The resulting constructs were injected into embryos of attP2 site with PhiC31-mediated transgenesis. The correct insertion was screened by *vermillion* (vermillion positive eye color) and verified by PCR and followed by DNA sequencing. Oligo primers are as following:
Oligo forward:
5′-ctagcagttTGGCTGTGAAGTGAAATTGTAtagttatattcaagcataTACAATTTC
ACTTCACAGCCAgcg-3′
Oligo reverse:
5′-aattcgcTCGCACTGTAGCCCAGATCTAtatgcttgaatataactaTACAATTTCA
CTTCACAGCCAactg-3′.

**Generation of UAS-flag-dsx^F^, UAS-myc-dsx^M^ and UAS-flag-dsx^M2^ lines**. *UAS-flag-dsx^F^* and *UAS-myc-dsx^M^* lines were generated previously in our lab[35]. The *UAS-flag-dsx^M2^* line was generated in this study. In brief, the 3×Flag tags (MDYKDHDG-DYKDHDI-DYKDDDDKL) were added to the N terminus of Dsx^F^ and Dsx^M2^ protein separately. We simultaneously amplified the *flag* sequence and the full-length CDS fragment of *dsx^F^* and *dsx^M2^* from wild-type cDNAs using

**Table 1 Detailed information for key resources and fly stocks.**

| Reagent type (species) or resource | Designation | Source or reference | Identifiers | Additional information |
|---|---|---|---|---|
| Antibody | Mouse monoclonal anti-Flag | Sigma–Aldrich | Cat# F1804 | IHC (1:500) |
| Antibody | Mouse monoclonal anti-Myc | MBL | M047-3 | IHC (1:200) |
| Antibody | Donkey polyclonal anti-Mouse, Alexa Fluor 488 | Thermo Fisher Scientific | Cat# A-21202, RRID: AB_141607 | IHC (1:500) |
| Antibody | Donkey polyclonal anti-Mouse, Alexa Fluor 555 | Thermo Fisher Scientific | Cat# A-31570, RRID: AB_2536180 | IHC (1:500) |
| Plasmid | pVALIUM20 | Tsinghua University | | |
| Plasmid | pJFRC2-10×UAS-IVS-mCD8::GFP | Addgene | #26214 | |
| Chemical compound drug | Normal Goat Serum (NGS) | Jackson ImmunoResearch Laboratories | Code# 005-000-121 RRID: AB_2336990 | 3% NGS in 1×PBS |
| Chemical compound drug | Paraformaldehyde (PFA) | Sigma–Aldrich | CAS# 30525-89-4 | 4% PFA in 1×PBS |
| Chemical compound drug | TRIzol™ reagent | Invitrogen | Cat# 15596026 | |
| Chemical compound drug | SuperScript™ IV | Invitrogen | Cat# 18091050 | |
| Chemical compound drug | DNA Polymerase High Fidelity | Transgen | Cat# AS131-21 | |
| Chemical compound drug | EvaGreen Dye | Biotium | Cat# 31000 | |
| Genetic reagent (D. melanogaster) | UAS-dsx$^M$RNAi | 14 | | |
| Genetic reagent (D. melanogaster) | dsx$^{GAL4}$ | 12 | | |
| Genetic reagent (D. melanogaster) | dsx$^{GAL4(\Delta 2)}$ | 36 | | |
| Genetic reagent (D. melanogaster) | R57C10-GAL4 | Bloomington Drosophila Stock Center | BDSC_39171 | |
| Genetic reagent (D. melanogaster) | actin-GAL4 | Bloomington Drosophila Stock Center | BDSC_25374 | |
| Genetic reagent (D. melanogaster) | UAS-traRNAi | Bloomington Drosophila Stock Center | BDSC_28512 | |
| Genetic reagent (D. simulans) | Drosophila simulans | 13 | | |
| Genetic reagent (D. mojavensis) | Drosophila mojavensis | 13 | | |
| Genetic reagent (D. virilis) | Drosophila virilis | 13 | | |
| Genetic reagent (D. melanogaster) | UAS-dsx$^{M2}$RNAi | This study | | Described below |
| Genetic reagent (D. melanogaster) | UAS-flag-dsx$^{M2}$ | This study | | Described below |
| Genetic reagent (D. melanogaster) | UAS-flag-dsx$^F$ | 35 | | Described below |
| Genetic reagent (D. melanogaster) | UAS-myc-dsx$^M$ | 35 | | Described below |
| Software,algorithm | LaserGene | DNAStar | http://www.dnastar.com/ | |
| Software,algorithm | Clustal Omega | EMBL-EBI | https://www.ebi.ac.uk/Tools/msa/clustalo | |
| Software,algorithm | ImageJ | ImageJ National Institutes of Health | https://imagej.nih.gov/ij/ | |
| Software,algorithm | Prism 8 | GraphPad | https://www.graphpad.com/ | |
| Software,algorithm | Integrative Genomics Viewer | 37 | https://www.igv.org/ | |

**Table 2 Primers used for RT-PCR experiments.**

| Usage | Primer names | Sequences (5′–3′) |
|---|---|---|
| Fig. 1d, e | Primer 1 | Forward: CAATCGCTGGAGGGGTCCTG<br>Reverse: TCATCCACATTGCCGCGTTG |
| Fig. 1d, e | Primer 2 | Forward: GCAAAGCACACCTCGCGGAG<br>Reverse: TCATCCACATTGCCGCGTTG |
| Fig. 1d, e | Primer 3 | Forward: TTCCGCTATCCTTGGGAGCT<br>Reverse: TTGGCTTGTATGCCTATTCG |
| Fig. 3a, b | Primer 1 | Forward: GAATCATGGTTTCGGAGGAGAACTG<br>Reverse: CTACGTGGCAGCCGTGGAGCTCACC |
| Fig. 3a, b | Primer 2 | Forward: GAATCATGGTTTCGGAGGAGAACTG<br>Reverse: TCATCCACATTGCCGCGTTGTGTTGC |
| Fig. 3d, e | Primer 1 | Forward: ATGGTTTCGGAGGAGAACTGG<br>Reverse: TCATCCACATTGCCGCGTTGT |
| Fig. 3d, e | Primer 2 | Forward: ACGCAAGAATGTGCCACTCG<br>Reverse: TCATCCACATTGCCGCGTTGT |
| Fig. 3g, h | Primer 1 | Forward: ATGGTTTCAGAGGAGAATTGGAACA<br>Reverse: TCATCCACATTGCCGCGTTGTGTTG |
| Fig. 3g, h | Primer 2 | Forward: AGCACACGCAAGAATGTGCCACTGG<br>Reverse: TCATCCACATTGCCGCGTTGTGTTG |
| Fig. 3j, k | Primer | Forward: AGAACGGCAGCGAGACAGGC<br>Reverse: TGTGGAGACGGGCGATGAGG |
| Supplementary Fig. 8b, c | Primer 1 | Forward: ATGGTTTCGGAGGAGAACTG<br>Reverse: TCATCCACATTGCCGCGTTG |
| Supplementary Fig. 8b, c | Primer 2 | Forward: CAATCGCTGGAGGGGTCCTG<br>Reverse: TCATCCACATTGCCGCGTTG |

the following overhang primers and then cloned into pJFRC2-10XUAS-IVS-mCD8::GFP plasmid (Addgene #26214) to replace the mCD8::GFP sequence.

*dsx^F*-forward: CCGAGATCTATGGACTACAAAGACCATGACGGTGATTATAAAGATCATGACATCGATTACAAGGATGACGATGACAAGCTTGTTTCGGAGGAGAACTGG

*dsx^F*-reverse: CCATCTAGATCATCCACATTGCCGCGTTG.

*dsx^M2*-forward: CCGAGATCTATGGACTACAAAGACCATGACGGTGATTATAAAGATCATGACATCGATTACAAGGATGACGATGACAAGCTTGTTTCGGAGGAGAACTGG

*dsx^M2*- reverse: CCATCTAGATCACTGATCAACAGACTTACC.

For the *UAS-myc-dsx^M* transgenic line, the 6×Myc tag (MEQKLISEEDL) was added to the N terminus of Dsx^M. We first cloned the 6×myc sequence into pJFRC2-10XUAS-IVS-mCD8::GFP between the *BglII* and the *XhoI* sites, and then amplified the full-length CDS of *dsx^M* and cloned into C terminus of 6×Myc to replace the mCD8::GFP sequences.

*myc*-F: CCGAGATCTTCCCATCGACTTAAAGCTATG

*myc*-R: CCACTCGAGACTAGTCTCAAGAGGCCTTGAGTTC

*dsx^M*-F: CCGACTAGTGTTTCGGAGGAGAACTGG

*dsx^M*-R: CCATCTAGACTACGTGGCAGCCGTGGAG

The constructed plasmids were injected and integrated into the attP40 sites on the second chromosome through phiC31 integrase mediated transgenesis. The correct strains were screened by *mini-white* (orange eyes) and verified by PCR and followed by DNA sequencing.

**RT-PCR analysis**. To acquire cDNA template, *Canton-S, w^1118* and other species including *D. simulans, D. mojavensis, D. virilis* and *Blattella germanica* were used as wild-type strains for RT-PCR. We obtained total RNA from approximate thirty adult females and/or males using a commercial TRizol™ reagent (15596026, Invitrogen, USA) and purified RNA with DNA-free™ kit (AM1906, AMbion) according to manufacturer's protocol. Purified RNA was finally resuspended in 60μL of DEPC-treated water. First strand cDNA was synthesized for each RNA sample using SuperScript™ IV reverse transcriptase (18091050, Invitrogen). *dsx*-specific primers are listed in Table 2 to detect different *dsx* transcripts.

**qPCR**. Total RNA extraction and first strand cDNA template were acquired as described above, except for the data in Supplementary Fig. 3, which used approximately 100 flies to obtain different tissues (head, thorax, abdomen, forelegs, and wings). qPCR was performed using a LightCycler® 96 SW 1.1 system (Roche). We used EvaGreen Dye (31000, Biotium, USA) and High Fidelity PCR SuperMix (AS131–21, TransGen, Beijing) to conduct qPCR. *actin* was amplified as an internal control for normalization. The primers for qPCR used in this study are listed in Table 3.

**Male courtship assay**. Newly enclosed males were collected and group housed for 4–7 days at 25°C and 60% humidity with a 12 hr: 12 hr light/dark cycle. Virgin *Canton-S* females were group-housed and aged under similar conditions. To

**Table 3 Primers used for qPCR experiments.**

| | Sequences (5′–3′) |
|---|---|
| *actin* | Forward: CAGGCGGTGCTTTCTCTCTA<br>Reverse: AGCTGTAACCGCGCTCAGTA |
| *dsx^M* | Forward: GAAGAGGCTTCCCGGCGAAT<br>Reverse: GGACAAATCTGTGTGAGCGG |
| *dsx^F* | Forward: TTCCGCTATCCTTGGGAGCT<br>Reverse: CATCCACATTGCCGCGTTGT |
| *dsx^M2* | Forward: GCCGATCTCAGTTTCCGTCA<br>Reverse: TCACTGATCAACAGACTTAC |
| *yp2* | Forward: TGGGTCAATCCACGTGAAGT<br>Reverse: ACAATGTAGCCCCTGATCTG |
| *yp3* | Forward: GAAGCCGACCAAGTGGCTGA<br>Reverse: TCCAGACGGGCACATTGCTC |

measure courtship, a male of each genotype and a *Canton-S* virgin female were loaded individually into two-layer chambers (diameter: 1 cm; height: 3 mm per layer) which were separated by a plastic transparent barrier until courtship test for 30 min. Courtship index (CI), which is the percentage of observation time a fly displayed any courtship step in 10 min, and percentages mated were counted based on successful copulation in 30 min.

**Tissue dissection, staining, and imaging**. Adult flies were reared at 25°C and aged for 4–7 days old, and were dissected in Schneider's insect medium (Thermo Fisher Scientific, Waltham, MA) and fixed in 4% paraformaldehyde in phosphate-buffered saline (PBS) for 20–30 min at room temperature (RT). Tissues were washed at least 4 times for 15–20 min with PAT3 (0.5% Triton X-100, 0.5% bovine serum albumin in PBS), and then blocked in 3% normal goat serum (NGS) for 60 min at RT. In Supplementary Fig. 5e and f, samples were incubated with anti-Flag mouse (Sigma, F1804, 1:500) antibody or anti-Myc mouse (MBL, M047-3, 1:200) antibody diluted in 3% NGS for overnight at 4 °C, then washed four times in PAT3, and incubated in secondary antibodies anti-mouse IgG conjugated to Alexa 488 (Invitrogen, A21202, 1:500) diluted in 3% NGS for 1–2 days at 4 °C. Tissues were then washed thoroughly in PAT3 and mounted for confocal imaging.

For visualizing morphological appearances of flies (Fig. 2f, g and Supplementary Fig. 8a), 4–7 days old flies were frozen at -80°C for 30 min. Fly forelegs and abdomens were dissected and imaged by a Nikon Shuttle pix P400RV stereoscopic microscope.

**Statistics and reproducibility**. All statistical analyses were performed using the Prism 8 (GraphPad software). Experimental flies and genetic controls were tested at the same condition, and data are collected from at least two independent experiments and are reproducible. The Mann-Whitney U test was used for pairwise comparisons. For comparing mating success, Chi-square tests were performed to compare two different groups at 30 min time point.

**Reporting summary**. Further information on research design is available in the Nature Research Reporting Summary linked to this article.

## Data availability

All data generated or analyzed during this study are included in the manuscript and its supplementary information files. Source data underlying figures are presented in Supplementary Data 1. Uncropped gel images are provided in Supplementary Fig. 9. All other relevant data supporting the findings of this study are available from the corresponding author upon reasonable request.

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

## Acknowledgements

We thank the Bloomington *Drosophila* Stock Center and the Tsinghua Fly Center for fly stocks, and Dr. Sheng Li and Nan Chen (South China Normal University) for generously providing cDNA of *Blattella germanica*. This work was supported by grants from National Key R&D Program of China (2019YFA0802400), the National Natural Science Foundation of China (31970943 and 31700905), and the Jiangsu Innovation and Entrepreneurship Team Program.

## Author contributions

C.H., Q.P. and Y.P. designed research; C.H., Q.P., X.S. and L.X. generated transgenic flies; C.H., and X.S. performed behavioral experiments; C.H. and X. Ji. performed immunostaining and qPCR experiments; C.H., Q.P. and Y.P. analyzed data; and Y.P. wrote the manuscript with input from all authors.

## Competing interests

The authors declare no competing interests.
