## [Peer Review File · Communications Biology]

Reviewers' comments:

Reviewer #1 (Remarks to the Author):

In this manuscript, Han and colleagues describe a novel doublesex male-specific isoform, dsxM2, that is produced by intron retention (IR). They characterize the effect of knockdown and overexpression of this isoform and find that it has masculinizing function that is less potent than the dsxM. They then show that there is wide conservation of this isoform in insects and propose that the dsxM2 isoform as the ancestral form of the transcript prior to sexual dimorphism in isoforms, through an intermediate intron retention only phase, followed by an IR and alternative polyadenylation phase (APA). The evidence for the existence of this isoform and the proposed evolutionary pathway are compelling and the writing is clear. I have some comments and concerns which, if addressed, would improve the quality of the manuscript.

Major comments:

- I urge the authors to use publically available RNAseq datasets to provide independent evidence of intron retention that is specific to males. In addition to that, it would be interesting to see if there is any tissue specificity in this isoform's expression.
- Is the DsxM2 lowly expressed compared to DsxM? It would be valuable to have an idea of the relative abundances of the isoforms (i.e. stoichiometry) in males. This may be achieved experimentally or bioinformatically using publically available RNAseq datasets.
- The interpretation of DsxM/M2 knockdown experiments (Figure 2a,c) are lacking a very important element: efficiency of knockdown. It is clear from supplementary figure 2b,c that the knockdown of DsxM is much more efficient than that of DsxM2. The authors do not discuss whether this reduced efficiency is affecting the results in figure 2. Related to that, it is not clear whether the efficiency is also different when driven pan-neuronally using R57C10-GAL4.
- In the overexpression experiments, it is not clear why dsxM2 and dsxF were constructed with a flag tag, while dsxM with a myc tag. This precludes any direct comparison between the levels of overexpression, and therefore interpretation of their differential phenotypic effects in figures 2f,g.

Minor comments:

- The proposed evolutionary pathway in figure 5c is very compelling but will require more work to confirm. The abstract on line 28, is too assertive and leads the reader to think that the proposed pathway is supported by evidence. I would suggest a slight rephrasing.
- Line 81: I believe authors want to say "difference in conservation" instead of "conserveness".
- Line 118: would be interesting to know more about why they think they have failed to produce antibodies specific to DsxM2.
- Lines 145,282: "masculinized" -> "masculinizing"
- Line 180: "conserveness" -> "levels of conservation"
- In many places in the manuscript: "conserveness" with "conservation"

Reviewer #2 (Remarks to the Author):

Han et al report the discovery of a new splicing isoform of the *D. melanogaster* dsx transcript. The transcript is related to dsx[F], but retains an additional intron. By RNAi and ectopic expression they show that this transcript, called dsx[M2] is male-determining, though potentially in a less powerful way than dsx[M]. They detect this splice isoform on three other *Drosophila* species and in the German cockroach. Based on this, and the level of conservation of sequences within dsx[M2] and dsx[F] relative to dsx[M], they propose that generation of dsx[M2] may have been an early step in the evolution of the sex-specific splice products of dsx.

The authors' identification of a new dsx splicing isoform is convincing, new, and interesting. The data were clear and the experiments appropriate. Nearly as convincing were the data showing that dsx[M2] has at least some male-determining function, but additional data are necessary to make it fully convincing, as detailed below. The idea that dsx[M2] is an intermediate along the path to evolve dsx[M] is intriguing, but finding it in three *Drosophila* species and one cockroach is not

sufficient for certainty that this was the evolutionary path.

The paper is well written, with a comprehensive and accurate introduction, and the figures are similarly high quality. I think it would be publishable with additional data to clarify the extent or strength of dsx[M2] function, and some toning down of the evolutionary certainty of the model (or adding more species, including at important nodes of the tree).

Major comments, by line number:

119 and following: please provide information as to the extent of knockdown. It is possible that the smaller effect seen upon knockdown of dsx[M2] relative to dsx[M] could be because the former was not knocked down as much.

119 and following: it is also important show that knockdown of dsx[M2] does not impact levels of dsx[M]. And vice versa. It is also difficult to know from the description of the knockdown plasmid.

134 and following: similarly, it is important to show that the overexpression levels of Dsx[M] and Dsx[M2] are the same, to be able to compare their effects quantitatively, as done here.

224-225; also 233 and following: these are interesting suggestions, but still quite speculative. In 239ff, for example, I don't think the ancestral situation is known.

244, 444: contradict what is said on lines 124 and 283 which says that dsx[M] does not mediate male differentiation in terms of sexual dimorphism.

254: there are not enough insects tested to make this strong of an evolutionary conclusion. Lines 257-258, 260-262, 267-268 are similarly overly speculative.

262-263: no data are provided to support this statement.

284: direct evidence could be provided by CRISPR/Cas9-mediated substitutions

Minor comments, by line number:

81, 180: conservation, rather than conserveness

102: from females, rather than from female ones

110: resulting, rather than resulted

115: it is a result of intron retention, in the sequence sense, but we do not know the mechanism

134: Are the authors certain that tags do not impair function?

215: remove "of" from despite of many

232, 242, 272: evolution, rather than evolvment

321: vermillion, rather than vermilion

389: change to: counted based on successful copulation in 30 min

Reviewer #3 (Remarks to the Author):

Identification of a novel dsx isoform reveals an evolutionary pathway of sexual development via distinct alternative splicing mechanisms

The doublesex/mab-3 related transcription factor (Dmrt) gene, double sex (dsx) in *Drosophila* produces sex specific dsxF and dsxM transcripts through alternative splicing with mutually exclusive exons resulting in a DNA binding domain with one of two effector domains, which act as repressors and possibly activators. The study identified a novel male-specific dsx transcript (dsxM2) generated through a well-known intron retention (IR) splicing mechanism. The intron retention leads to a dsxM2 transcript encoding an extremely truncated effector domain. The first main observation is that this male isoform that is conserved and therefore likely to be important. The RNAi work on the novel isoform shows a subtle effect on male behavior. This is the 2nd novel point in the paper. The results are straight-forward and the work well done, although I do have some problems with over-interpretation in the results (I think authors should be free to speculate in the discussion).

Major:

1. Although not the main point of any papers, the intron retention the authors report is known in the field and is obvious in the read density maps that have been generated in previous work in

Drosophila. The authors over-state the novelty of this underlying observation.

2. The authors stress that dsxM2 masculinizes the flies similar to dsxM and is therefore a critical isoform for males. They also stress that male-specific intron retention depends on the presence of a weak splicing acceptor sequence inside the intron, and regulation by Tra. I am not convinced by either of these arguments.

3. The Dsx locus has long been thought to encode cross-repression as a mechanism, so over-expressing the DNA binding domain of DsxM2 in females and generating a morphological or yolk protein expression phenotype is not as informative as it might seem. DsxM2 might have no role in these processes, but acts as a sort of dominant negative when expressed in females. So, I disagree with the statement that masculinization of females indicates that DsxM2 has a role in masculinizing males; in fact the DsxM2 RNAi suggests that there is no morphological masculinizing effect of DsxM2 in males.

4. The idea that there is a weak tra enhancer site was not tested. Given how difficult these sites are to identify, it is speculation.

Minor:

1. In Fig.1c, an arrowhead could point to the dsxM2 transcript as this is the first time dsxM2 appears in the paper. In the legend, the arrowhead information could be added to the existing lines 433-435.

2. In Fig.1e, distinguish the length of the amplified products in the text/legend as later the products can be related to dsxF or dsxM2 based on differences in size.

3. The Model in Fig.5 is confusing. 5a initial part is redundant with Fig.5c in showing sex-specific dsx transcripts were thought to be evolved from sexually monomorphic dsx with male-biased expression. 5d is not necessary and maybe inaccurate as not 'all' the dsxM regulation might be through repression. The model could be one/two piece that includes the bottom part of 5 a/b with c.

Our point-to-point reply to comments are in blue.

Reviewer #1 (Remarks to the Author):

In this manuscript, Han and colleagues describe a novel doublesex male-specific isoform, *dsxM2*, that is produced by intron retention (IR). They characterize the effect of knockdown and overexpression of this isoform and find that it has masculinizing function that is less potent than the *dsxM*. They then show that there is wide conservation of this isoform in insects and propose that the *dsxM2* isoform as the ancestral form of the transcript prior to sexual dimorphism in isoforms, through an intermediate intron retention only phase, followed by an IR and alternative polyadenylation phase (APA). The evidence for the existence of this isoform and the proposed evolutionary pathway are compelling and the writing is clear. I have some comments and concerns which, if addressed, would improve the quality of the manuscript.

Reply: We thank the reviewer for the detailed summary and thoughtful comments.

Major comments:

- I urge the authors to use publically available RNAseq datasets to provide independent evidence of intron retention that is specific to males. In addition to that, it would be interesting to see if there is any tissue specificity in this isoform's expression.

Reply: We thank the reviewer for this suggestion. We have checked the supplementary data from Brown et al., 2014, Nature (Diversity and dynamics of the Drosophila transcriptome). Indeed, Brown et al., identified 9 transcripts and 3 proteins form Drosophila *dsx* (from their supplementary table 3 that shows the number of transcripts and proteins of genes, but without detailed information). We further checked the modENCODE database (<http://intermine.modencode.org/release-33/begin.do>), which is related to the above reference (Brown et al., 2014, Nature), but only found 6 transcripts and 2 proteins from *dsx* in this database, which is the same as found in flybase (<https://flybase.org/>). Nevertheless, we believe our results adequately indicate the existence of the novel *dsx^{M2}* transcript.

To address if there is any tissue specificity of the *dsx^{M2}* transcript, we performed qPCR experiments against *dsx^M* and *dsx^{M2}* in various tissues (head, thorax, abdomen, wings, and forelegs) and found that the expression of *dsx^M* is generally higher than *dsx^{M2}*, and the *dsx^{M2}* transcript is preferentially expressed in the head, thorax, and forelegs, but weakly in the abdomen and wings. This new result is now added as Supplementary Fig. 1.

- Is the *DsxM2* lowly expressed compared to *DsxM*? It would be valuable to have an idea of the relative abundances of the isoforms (i.e. stoichiometry) in males. This may

be achieved experimentally or bioinformatically using publically available RNAseq datasets.

Reply: We thank the reviewer for this valuable suggestion and re-analyzed our previous qPCR data in Fig. 1b and c. We found that dsx^{M2} is indeed expressed in a lower level compared to dsx^M (~40%) in males. Thus, we used this new analysis and updated Fig. 1. In addition, qPCR experiments using different tissues also revealed that expression of dsx^{M2} is generally lower than dsx^M in all tested tissues (Supplementary Fig. 1).

- The interpretation of DsxM/M2 knockdown experiments (Figure 2a,c) are lacking a very important element: efficiency of knockdown. It is clear from supplementary figure 2b,c that the knockdown of DsxM is much more efficient than that of DsxM2. The authors do not discuss whether this reduced efficiency is affecting the results in figure 2. Related to that, it is not clear whether the efficiency is also different when driven pan-neuronally using R57C10-GAL4.

Reply: We now repeated the qPCR experiments to further validate the efficiency of dsx^M and dsx^{M2} RNAi lines using three *GAL4* drives: *actin-GAL4*, *R57C10-GAL4* and *dsx^{GAL4}*. We found that all three *GAL4* lines driving dsx^M RNAi significantly reduced dsx^M expression, with the *actin-GAL4* and *dsx^{GAL4}* more efficient than the *R57C10-GAL4* driver. We also found that all three *GAL4* lines driving dsx^{M2} RNAi significantly reduced dsx^{M2} expression (reduced by ~50% with the *actin-GAL4*), but not as strong as using the dsx^M RNAi (reduced by ~80% with the *actin-GAL4*).

Thus, both RNAi lines work efficiently but may not knock down corresponding transcripts in the same level. We have discussed such a caveat in the discussion section in the revised manuscript. The new qPCR data was added as a part of the new Fig. 2.

- In the overexpression experiments, it is not clear why dsxM2 and dsxF were constructed with a flag tag, while dsxM with a myc tag. This precludes any direct comparison between the levels of overexpression, and therefore interpretation of their differential phenotypic effects in figures 2f,g.

Reply: We thank the reviewer for this comment. Indeed, it is possible that different tags would lead to different overexpression levels. We originally made such design as we would like to co-express different isoforms (e.g., Dsx^F and Dsx^M) in our other projects, and the use of different tags would allow us to distinguish these isoforms in the same fly (e.g., simultaneously using anti-Flag and anti-Myc antibodies).

We agree with the reviewer that there is a caveat that Dsx^M and Dsx^{M2} overexpression levels might be different, and we have discussed such a caveat in the revised manuscript. An example of textual changes in the discussion section as following: However, a direct comparison of Dsx^M and Dsx^{M2} function could not be faithfully achieved as knockdown and overexpression levels of dsx^M and dsx^{M2} may be different.

Minor comments:

- The proposed evolutionary pathway in figure 5c is very compelling but will require more work to confirm. The abstract on line 28, is too assertive and leads the reader to think that the proposed pathway is supported by evidence. I would suggest a slight rephrasing.

Reply: We agree with the reviewer and now tone down the evolutionary pathway in the revised manuscript.

- Line 81: I believe authors want to say "difference in conservation" instead of "conserveness".

Reply: Corrected.

- Line 118: would be interesting to know more about why they think they have failed to produce antibodies specific to DsxM2.

Reply: We rephrased the sentence as following: we tried to generate a polyclonal antibody against the eight amino acids in the C-terminus of the predicted Dsx^{M2} protein but failed to detect any signal through immunostaining and western blot experiments.

The failure of this antibody is further confirmed by our unpublished data with overexpression of Flag-Dsx^{M2} in *dsx^{GAL4}* cells, which could be successfully stained by anti-Flag but not anti-Dsx^{M2}.

- Lines 145,282: "masculinized" -> "masculinizing"

Reply: Corrected.

- Line 180: "conserveness" -> "levels of conservation"

Reply: Corrected.

- In many places in the manuscript: "conserveness" with "conservation"

Reply: We thank the reviewer for pointing this out and carefully checked this issue throughout the manuscript.

Reviewer #2 (Remarks to the Author):

Han et al report the discovery of a new splicing isoform of the *D. melanogaster* *dsx* transcript. The transcript is related to *dsx*[F], but retains an additional intron. By RNAi and ectopic expression they show that this transcript, called *dsx*[M2] is male-determining, though potentially in a less powerful way than *dsx*[M]. They detect this splice isoform on three other *Drosophila* species and in the German cockroach. Based on this, and the level of conservation of sequences within *dsx*[M2] and *dsx*[F] relative to *dsx*[M], they propose that generation of *dsx*[M2] may have been an early step in the evolution of the sex-specific splice products of *dsx*.

The authors' identification of a new *dsx* splicing isoform is convincing, new, and

interesting. The data were clear and the experiments appropriate. Nearly as convincing were the data showing that *dsx*[M2] has at least some male-determining function, but additional data are necessary to make it fully convincing, as detailed below. The idea that *dsx*[M2] is an intermediate along the path to evolve *dsx*[M] is intriguing, but finding it in three *Drosophila* species and one cockroach is not sufficient for certainty that this was the evolutionary path.

The paper is well written, with a comprehensive and accurate introduction, and the figures are similarly high quality. I think it would be publishable with additional data to clarify the extent or strength of *dsx*[M2] function, and some toning down of the evolutionary certainty of the model (or adding more species, including at important nodes of the tree).

Reply: We thank the reviewer for the comprehensive summary and these positive comments. We toned down the evolutionary pathway as suggested in the revised manuscript.

Major comments, by line number:

119 and following: please provide information as to the extent of knockdown. It is possible that the smaller effect seen upon knockdown of *dsx*[M2] relative to *dsx*[M] could be because the former was not knocked down as much.

Reply: We now repeated the qPCR experiments to further validate the efficiency of *dsx*^M and *dsx*^{M2} RNAi lines using three *GAL4* drives, *actin-GAL4*, *R57C10-GAL4* and *dsx*^{GAL4}. We found that all three *GAL4* lines driving *dsx*^M RNAi significantly reduced *dsx*^M expression, with the *actin-GAL4* and *dsx*^{GAL4} more efficient than the *R57C10-GAL4* driver. We also found that all three *GAL4* lines driving *dsx*^{M2} RNAi significantly reduced *dsx*^{M2} expression (reduced by ~50% with the *actin-GAL4*), but not as strong as using the *dsx*^M RNAi (reduced by ~80% with the *actin-GAL4*). Thus, both RNAi lines work efficiently but may not knock down corresponding transcripts in the same level. The new qPCR data was added as a part of the new Fig. 2.

We have discussed such a caveat in the discussion section in the revised manuscript (see below).

119 and following: it is also important show that knockdown of *dsx*[M2] does not impact levels of *dsx*[M]. And vice versa. It is also difficult to know from the description of the knockdown plasmid.

Reply: We thank the reviewer for this suggestion and now performed qPCR experiments to quantify expression of *dsx*^M and *dsx*^{M2} in each RNAi-mediated line. Indeed, we found that *dsx*^M or *dsx*^{M2} RNAi is efficient and specific to knock down the corresponding *dsx* transcript but not the other one. This new result is added as Supplementary Fig. 3.

134 and following: similarly, it is important to show that the overexpression levels of *Dsx*[M] and *Dsx*[M2] are the same, to be able to compare their effects quantitatively, as done here.

Reply: In addition to the immunostaining experiments with anti-Myc and anti-Flag, which indicates successful overexpression of Dsx isoforms, we now performed additional qPCR experiments to further validate the efficiency of the three overexpression constructs. We found that overexpression of each *dsx* transcript, driven by the *dsx^{GAL4}* in either males or females, dramatically increased the corresponding *dsx* transcript.

For overexpression of *dsx^M*, the level of *dsx^M* mRNA is increased from undetectable to ~7 (relative to *actin* mRNA, see Supplementary Fig. 4b-d) in females, and ~0.07 to ~6 in males; for overexpression of *dsx^{M2}*, the level of *dsx^{M2}* mRNA is increased from undetectable to ~9 in females, and ~0.04 to ~16 in males. As different primers were used to quantify *dsx^M* and *dsx^{M2}* mRNA levels, we could not make a solid conclusion by directly comparing their expression levels.

Thus, our immunostaining and qPCR experiments faithfully indicate the effective overexpression of *dsx^M* and *dsx^{M2}*, but a caveat that their expression levels might be different still exists and has been brought out in the discussion in the revised manuscript. An example of textual changes in the discussion section as following: However, a direct comparison of Dsx^M and Dsx^{M2} function could not be faithfully achieved as knockdown and overexpression levels of *dsx^M* and *dsx^{M2}* may be different.

224-225; also 233 and following: these are interesting suggestions, but still quite speculative. In 239ff, for example, I don't think the ancestral situation is known.

Reply: We have toned down this conclusion and used "suggest" instead of "indicate" in this and some other sentences in the revised manuscript.

244, 444: contradict what is said on lines 124 and 283 which says that dsx[M] does not mediate male differentiation in terms of sexual dimorphism.

Reply: We are sorry for this misunderstanding. *dsx^M* surely mediates male differentiation. As for *dsx^{M2}*, our knockdown experiments indicate its crucial role in the nervous system for male courtship robustness but not sexual development, while our overexpression experiments indicate that Dsx^{M2} has a potentially masculinizing role like Dsx^M. We have rephrased these sentences to make them clearer to understand.

254: there are not enough insects tested to make this strong of an evolutionary conclusion. Lines 257-258, 260-262, 267-268 are similarly overly speculative.

Reply: We have toned down the evolutionary pathway in the revised manuscript.

262-263: no data are provided to support this statement.

Reply: This statement is supported by the Supplementary Fig. 7, which showed that the *dsx^{M2}* transcript was yielded by knocking down *tra* in females. We also toned down this conclusion in the result section of the revised manuscript.

284: direct evidence could be provided by CRISPR/Cas9-mediated substitutions

Reply: We thank the reviewer for suggestion of this very important experiments. Indeed, CRISPR/Cas9-mediated knock-in would help to reveal where dsx^{M2} expresses, which would certainly enhance the manuscript.

We believe the current manuscript, without knowing the Dsx^{M2} expression data (even if Dsx^{M2} has very limited expression and function), stands well regarding to the identification of the dsx^{M2} transcript and suggestion of an evolutionary pathway. An example of textual changes in the discussion section as following: It is also possible that Dsx^{M2} only serves as an ancient Dsx^M isoform whose function has largely been replaced during evolution (*e.g.*, by Dsx^M), and now has limited expression and function.

Thus, we hope the reviewer could agree with us that this experiment, which is surely helpful but not crucial, requires substantially more time in the currently difficult situation, and we would like to leave it for future investigation (as we pointed out in the end of the discussion section).

Minor comments, by line number:

81, 180: conservation, rather than conserveness

Reply: Corrected.

102: from females, rather than from female ones

Reply: Corrected.

110: resulting, rather than resulted

Reply: Corrected.

115: it is a result of intron retention, in the sequence sense, but we do not know the mechanism

Reply: We rephrased the sentence as following: The above results identified a novel male-specific dsx^{M2} transcript, but how it is generated from intron retention and whether it plays any role in sexual development or behavior is not known.

134: Are the authors certain that tags do not impair function?

Reply: Indeed, we are not sure about this. We made such design to use the anti-Myc and anti-Flag antibodies for immunostaining or western blotting in this and other projects related to Dsx function, and so far, these reagents work very well.

215: remove “of” from despite of many

Reply: Corrected.

232, 242, 272: evolution, rather than evolvement

Reply: We have made the suggested change throughout the manuscript.

321: vermillion, rather than vermilion

Reply: Corrected.

389: change to: counted based on successful copulation in 30 min

Reply: Corrected.

Reviewer #3 (Remarks to the Author):

Identification of a novel dsx isoform reveals an evolutionary pathway of sexual development via distinct alternative splicing mechanisms

The doublesex/mab-3 related transcription factor (Dmrt) gene, double sex (dsx) in *Drosophila* produces sex specific dsxF and dsxM transcripts through alternative splicing with mutually exclusive exons resulting in a DNA binding domain with one of two effector domains, which act as repressors and possibly activators. The study identified a novel male-specific dsx transcript (dsxM2) generated through a well-known intron retention (IR) splicing mechanism. The intron retention leads to a dsxM2 transcript encoding an extremely truncated effector domain. The first main observation is that this male isoform is conserved and therefore likely to be important. The RNAi work on the novel isoform shows a subtle effect on male behavior. This is the 2nd novel point in the paper. The results are straight-forward and the work well done, although I do have some problems with over-interpretation in the results (I think authors should be free to speculate in the discussion).

Reply: We thank the reviewer for the comprehensive summary and very positive comments. As suggested, we now toned down the evolutionary pathway in the revised manuscript.

Major:

1. Although not the main point of any papers, the intron retention the authors report is known in the field and is obvious in the read density maps that have been generated in previous work in *Drosophila*. The authors over-state the novelty of this underlying observation.

Reply: We agree with the reviewer and were not intended to emphasize intron retention as a novel finding in this manuscript. We have made textual changes in this part (with track changes).

2. The authors stress that dsxM2 masculinizes the flies similar to dsxM and is therefore a critical isoform for males. They also stress that male-specific intron retention depends on the presence of a weak splicing acceptor sequence inside the intron, and regulation by Tra. I am not convinced by either of these arguments.

Reply: We thank the reviewer for this comment. Regarding to dsx^{M2} function, we made textual changes to better describe the results. An example in the discussion: Our knockdown experiments reveal that Dsx^{M2} still functions in the nervous system to promote courtship robustness in *D. melanogaster*, but not in the development of

male-specific traits such as genitals and sex combs. In addition, the overexpression experiments indicate that Dsx^{M2} has a potential masculinizing role just like Dsx^M .

As for the latter, we toned down the conclusion as following: these results suggest that the weak splice acceptor and *tra* regulation jointly determine the splicing modes between dsx^{M2} (intron retained) and dsx^F (intron spliced out).

3. The *Dsx* locus has long been thought to encode cross-repression as a mechanism, so over-expressing the DNA binding domain of *DsxM2* in females and generating a morphological or yolk protein expression phenotype is not as informative as it might seem. *DsxM2* might have no role in these processes, but acts as a sort of dominant negative when expressed in females. So, I disagree with the statement that masculinization of females indicates that *DsxM2* has a role in masculinizing males; in fact the *DsxM2* RNAi suggests that there is no morphological masculinizing effect of *DsxM2* in males.

Reply: We understand the concern of the reviewer. Indeed, overexpressing Dsx^{M2} masculinized females, which is perhaps due to the function of the common *Dsx* DNA binding domain interfering with the endogenous Dsx^F protein, as previously shown for the Dsx^M and Dsx^F proteins. That's why we also performed overexpression experiments under a *dsx* mutant background (Fig. 2f in wild-type background, and Fig. 2g in *dsx* mutant background). As overexpression of Dsx^{M2} masculinized both *dsx* mutant males and females (strong evidence by the sex combs), we believe it is reasonable to conclude that Dsx^{M2} has a potentially masculinizing role.

We discussed potential function of Dsx^{M2} in the last paragraph of the discussion section and pointed out potential caveats of this study.

4. The idea that there is a weak *tra* enhancer site was not tested. Given how difficult these sites are to identify, it is speculation.

Reply: We make textual changes as following: The conservation of the above core sequence, which is a potential *tra* binding site within the retained introns, suggests its potential role in *tra*-mediated alternative splicing.

Minor:

1. In Fig.1c, an arrowhead could point to the *dsxM2* transcript as this is the first time *dsxM2* appears in the paper. In the legend, the arrowhead information could be added to the existing lines 433-435.

Reply: We thank the reviewer for this very helpful suggestion and did so in the revised manuscript.

2. In Fig.1e, distinguish the length of the amplified products in the text/legend as later the products can be related to *dsxF* or *dsxM2* based on differences in size.

Reply: We added such description in the figure legend as suggested.

3. The Model in Fig.5 is confusing. 5a initial part is redundant with Fig.5c in showing

sex-specific dsx transcripts were thought to be evolved from sexually monomorphic dsx with male-biased expression. 5d is not necessary and maybe inaccurate as not 'all' the dsxM regulation might be through repression. The model could be one/two piece that includes the bottom part of 5 a/b with c.

Reply: We thank the reviewer very much for this comment and revised the model as suggested.

Reviewers' comments:

Reviewer #1 (Remarks to the Author):

I thank the authors who have adequately addressed the majority of my concerns. I am, however, still confused by their response to my first request about using public RNAseq datasets to confirm the presence of the intron retention. I will try to be more detailed:

1- It is still not clear to me how the already known isoforms, which are different in different annotations (in Flybase, modENCODE, or Brown et al., 2014) compare to the novel dsxM2 isoform. Can the authors confirm that none of these isoforms includes the retained intron? It would be a good idea to have all these isoforms in a supplementary figure, including their IDs, with the intron of interest highlighted.

2- The public datasets can be used to check whether read density in RNAseq alignments (using IGV for example) are consistent with intron retention producing dsxM2. Datasets from males and females can be used to show whether males have more retention than females. Sashimi plots from males and females would be useful, but not crucial, to illustrate that.

Reviewer #2 (Remarks to the Author):

I was reviewer 2 of the original manuscript.

The authors revised the manuscript to attempt to address the reviews of myself and the other two experts, and that revised manuscript is improved over the original. All of my minor comments were addressed satisfactorily, and some but not all of the major ones were too. For example the authors now show the important control that knocking down Dsx[M] does not affect Dsx[M2] and vice versa. Some other major comments were addressed by rewording.

However, the responses to some of my comments were not fully satisfying.

First, it was good to see the authors add the requested quantitation about extent of knockdown, but the knockdown extent for Dsx[M2] is poor; 50% is what you would see for one allele/deletion. For many genes that is not enough to conclude anything about phenotype.

Similarly, the data on quantifying overexpression provided more than the original submission, but still not enough to compare the overexpression of the two isoforms, which is what I and at least one other reviewer requested.

Third, though the change to "suggest" is a move in the right direction, the authors do not tone down enough their evolutionary conclusions. For example, in line 204 they have the section heading "Intron retention is a conserved mechanism to generate...". They have only looked at 4 Drosophila species and a cockroach. They do not know whether it is a conserved mechanism or whether it evolved independently in some lineages. It would be OK to say "common"; even better to simply state what they saw: IR was seen in three other Drosophila species and a cockroach. Similarly, in line 282 the heading should be "A proposed evolutionary pathway..." Otherwise it sounds like they have demonstrated it, which they haven't. They also still need to go through the paper and tone down the evolutionary conclusion.

While I think the CRISPR experiment would solve the problems, I understand that it may require more than the authors can do at this point.

Reviewer #3 (Remarks to the Author):

The authors have been responsive and make the distinction between data and speculation.

Our point-to-point reply to comments are in blue.

Reviewer #1 (Remarks to the Author):

I thank the authors who have adequately addressed the majority of my concerns. I am, however, still confused by their response to my first request about using public RNAseq datasets to confirm the presence of the intron retention. I will try to be more detailed:

1- It is still not clear to me how the already known isoforms, which are different in different annotations (in Flybase, modENCODE, or Brown et al., 2014) compare to the novel *dsx*^{M2} isoform. Can the authors confirm that none of these isoforms includes the retained intron? It would be a good idea to have all these isoforms in a supplementary figure, including their IDs, with the intron of interest highlighted.

Reply: We thank the reviewer for this very useful suggestion. There are six *dsx* transcripts in Flybase and modENCOD databases with consistent annotations. We illustrated the structures of the six *dsx* transcripts with their IDs and highlighted the 114bp intron, which is spliced out in female transcripts and retained in the new *dsx*^{M2} transcript. See the new Supplementary Fig. 1a below the next comment.

2- The public datasets can be used to check whether read density in RNAseq alignments (using IGV for example) are consistent with intron retention producing *dsx*^{M2}. Datasets from males and females can be used to show whether males have more retention than females. Sashimi plots from males and females would be useful, but not crucial, to illustrate that.

Reply: We thank the reviewer for this very straightforward suggestion and now utilized the RNAseq data from Graveley et al., (The developmental transcriptome of *Drosophila melanogaster*, Nature, 2011). Indeed, we found further support from this database that the 114bp intron is likely to be retained in males but not in females. See the new Supplementary Fig. 1 and Table 1 below.

Supplementary Fig. 1: Overview of *dsx* transcripts in *D. melanogaster* from public datasets. **a** Illustration of the six *dsx* transcripts based on the Flybase and modENCODE databases in *D. melanogaster*. **b** Sashimi plot visualization of *dsx* splicing events in *D. melanogaster* based on public datasets (<https://www.ncbi.nlm.nih.gov/geo/query/acc.cgi?acc=GSE28078>). The black dotted box indicates a zoomed region that includes the 114 bp intron (arrow), which is

spliced out in female samples but likely to be retained in male samples. For detailed figure description, see Supplementary Information-Figure 1.

In addition, we also found supporting evidence from the Table S27 of Graveley et al. (The developmental transcriptome of *Drosophila melanogaster*, Nature, 2011), which also stated that the 114 bp intron retention occurs in males but not females (see the below Table based on line 25068 of the Table S27).

as_event_type	intron_retention
gene_name	dsx
chr	chr3R
intron-exon_junctions	chr3R_3761375_3761376; chr3R_3761489_3761490
neighboring_constitutive_exons	chr3R_3760200_3761375; chr3R_3761490_3761627
0-24 hr embryos	NA
L1 larvae	NA
L2 larvae	NA
L3 larvae	NA
WPP + 12 hr	12.9
WPP + 24 hr	20
pupae, WPP + 2 days	24.49
pupae, WPP + 3 days	NA
pupae, WPP + 4 days	NA
adult male, eclosion + 1 day	94.74
adult male, eclosion + 5 days	85.94
adult male, eclosion + 30 days	93.94
adult female, eclosion + 1 day	0
adult female, eclosion + 5 days	NA
adult female, eclosion + 30 days	NA

Supplementary Table. 1 Public RNA-seq data shows the 114 bp intron retention in males but not females.

Reviewer #2 (Remarks to the Author):

I was reviewer 2 of the original manuscript.

The authors revised the manuscript to attempt to address the reviews of myself and the other two experts, and that revised manuscript is improved over the original. All of my minor comments were addressed satisfactorily, and some but not all of the major ones were too. For example the authors now show the important control that knocking down Dsx[M] does not affect Dsx[M2] and vice versa. Some other major comments were addressed by rewording.

However, the responses to some of my comments were not fully satisfying.

First, it was good to see the authors add the requested quantitation about extent of knockdown, but the knockdown extent for Dsx[M2] is poor; 50% is what you would see for one allele/deletion. For many genes that is not enough to conclude anything about phenotype.

Reply: We thank the reviewer for this comment. Indeed, the *dsx*^{M2} knockdown efficiency is ~50% and not as strong as *dsx*^M knockdown, but this would not compromise the main findings in this manuscript, which identifies the novel *dsx*^{M2} transcript through intron retention and suggests an evolutionary pathway of sex-specific splicing of *dsx*.

Dsx^{M2} could only serve a limited function, as we discussed in the manuscript "It is also possible that Dsx^{M2} only serves as an ancient Dsx isoform whose function has largely been replaced during evolution (e.g., by Dsx^M), and now has limited expression and function."

Similarly, the data on quantifying overexpression provided more than the original submission, but still not enough to compare the overexpression of the two isoforms, which is what I and at least one other reviewer requested.

Reply: We thank the reviewer for this comment. We proved that these overexpression lines work efficiently, but it is indeed not very conclusive to compare the efficiency of different transgenic lines on different transcripts using different primers for qPCR. We hope the reviewer could agree with us that such comparison is not essential. We are aware of such a caveat and stated clearly in the Discussion "However, a direct comparison of Dsx^M and Dsx^{M2} function could not be faithfully achieved as knockdown and overexpression levels of *dsx*^M and *dsx*^{M2} may be different."

Third, though the change to "suggest" is a move in the right direction, the authors do not tone down enough their evolutionary conclusions. For example, in line 204 they have the section heading "Intron retention is a conserved mechanism to generate...". They have only looked at 4 Drosophila species and a cockroach. They do not know whether it is a conserved mechanism or whether it evolved independently in some lineages. It would be OK to say "common"; even better to simply state what they saw: IR was seen in three other Drosophila species and a cockroach. Similarly, in line 282 the heading should be "A proposed evolutionary pathway..." Otherwise it sounds like

they have demonstrated it, which they haven't. They also still need to go through the paper and tone down the evolutionary conclusion.

Reply: We agree with the reviewer and have carefully gone through the manuscript to make conclusions more precisely. These textual changes are highlighted in the track-change mode. We thank the reviewer very much for these comments.

While I think the CRISPR experiment would solve the problems, I understand that it may require more than the authors can do at this point.

Reply: We thank the reviewer for the understanding and support.

REVIEWERS' COMMENTS:

Reviewer #1 (Remarks to the Author):

The authors have addressed my remaining comments and included the required public information (from high throughput sequencing) that support their findings.

Reviewer #1 (Remarks to the Author):

The authors have addressed my remaining comments and included the required public information (from high throughput sequencing) that support their findings.

Reply: We thank the reviewer for all the constructive comments during the whole review process.